# Study on Water Resource Carrying Capacity of Zhengzhou City Based on DPSIR Model

**DOI:** 10.3390/ijerph20021394

**Published:** 2023-01-12

**Authors:** Yizhen Jia, Han Wang

**Affiliations:** 1School of Business, Jiangnan University, Wuxi 214122, China; 2School of Accounting, Wuxi Taihu University, Wuxi 214063, China

**Keywords:** DPSIR model, water resources carrying capacity, combination weighting method, TOPSIS model

## Abstract

Based on the driving force–pressure–state–impact–response (DPSIR) model, a comprehensive evaluation index system is constructed. The index weight is determined by the combination weighting method in combination with the data of 2010–2019. The TOPSIS model is used to comprehensively analyze the water resource carrying capacity of Zhengzhou as the central city in China with a developed economy and relatively short water resources. The study results are as follows. (1) During the sample period, the comprehensive evaluation value of water resources carrying capacity of Zhengzhou increases from 0.4183 in 2010 to 0.5560 in 2019, with an overall fluctuating rise. Simultaneously, the water resource carrying capacity grade improves from Grade III (normal carrying capacity) to Grade II (good carrying capacity). (2) The contribution of each subsystem to the comprehensive evaluation value increases year by year. Among them, S subsystem and I subsystem make the largest contribution to the comprehensive carrying capacity. R subsystem makes a relatively stable contribution to the overall carrying capacity. Affected by GDP growth rate and uneven temporal–spatial distribution of water resources in Zhengzhou, the D subsystem and P subsystem of water resource carrying capacities show the fluctuating change. Finally, based on the above conclusions, this paper puts forward the countermeasures and suggestions to improve the level of water resource carrying capacity of Zhengzhou.

## 1. Introduction

Water as a resource is indispensable to people’s lives. With the growth of population and rapid development of economy in China, water consumption gradually increases. However, the pollution and shortage of water resources are increasingly severe, thus greatly affecting sustainable development and the utilization of water resources. Water resource carrying capacity can reflect the change trend of water resources and social development condition. As such, a study on water resource carrying capacity can provide guidance for rational regulation and sustainable utilization of water resources, as well as effectively deal with the relationship between water resources and the social economy. Hence, it is of great significance to study water resource carrying capacity for promoting regional sustainable development [1,2,3]. According to the previous literature, the earliest research on water resource carrying capacity began in the 1970s. The Organization for Economic Co-operation and Development (OECD) and the United Nations Educational Scientific and Cultural Organization (UNESCO) carried out a series of studies on the carrying capacity of natural resources such as forest, land and water in resource-deficient countries, and also put forward the concept of water resource carrying capacity for the first time [4,5,6]. Subsequently, scholars at home and abroad have deeply studied water resource carrying capacity from the perspectives of theory, method model and application. Among them, Swirepik [7] used the UEA evaluation method to predict the total demand for the eco-environmental water in eastern Australia and provided a reference for the sustainable diffluent management of water resources in the local whole basin. Likewise, Al-Kalbani [8] comprehensively evaluated water resource carrying capacity in the Sultanate of Oman with the DPSIR framework, as well as the sustainable utilization of local water resources. Based on the principal component analysis method, Cao Lijuan et al. [9] comprehensively evaluated and studied the water resource carrying capacity in some counties and cities of Gansu Province. Feng Xin et al. [10] reported that regional water resource carrying capacity is fuzzy and uncertain to a certain extent, so that the fuzzy mathematics method should be adopted for evaluation. In addition, Xu Kaili [11] used a system dynamics method to study the influence of water resource carrying capacity on the development of Lanzhou City, and then put forward a policy suggestion to construct a water-saving city with economic development.

To sum up, scholars at home and abroad have accumulated rich findings and experience on water resource carrying capacity, and they provide reference for further research. However, there are still a few issues to be solved until now. (1) The research region mostly focuses on the arid and semi-arid area. However, the central region, with more people and rapid development of economy, is less studied. (2) At present, the principal component analysis and fuzzy comprehensive evaluation methods are mainly used for research, but they cannot accurately reflect the difference between the actual water resource carrying capacity and the ideal value. (3) Major research efforts have been devoted to the analysis of water resource carrying capacity in one year but ignore the dynamic changes between different years. Moreover, the evaluation index weight is mostly determined by subjective weight method, failing in objectively reflecting the index weight.

Therefore, this paper selects Zhengzhou City, with rapid economy development and insufficient water resources, as the research region. An evaluation system is constructed based on DPSIR model, and the index weight is determined by combination weighting method. The TOPSIS model is used to comprehensively analyze the dynamic change trend of water resources carrying capacity of Zhengzhou from 2010 to 2019. Compared with previous studies, not only is the index system comprehensively constructed, but the difference between the current carrying capacity level and the ideal carrying capacity level is also directly reflected. Therefore, our results can objectively reflect the water resources carrying capacity and provide a reference for coordinating the relationships among water resources, ecological environment, as well as economic and social development in Zhengzhou.

## 2. Materials and Methods

### 2.1. Research Region

As the provincial capital of Henan, Zhengzhou is located in the center of China, downstream of the Yellow River and southern North China Plain. Thus, it is an important comprehensive transportation hub and central city in China. With a total area of 70,446 square kilometers, the terrain of Zhengzhou is high in the southwest and low in the northeast, dominated by plain and hill. Zhengzhou has jurisdiction over 12 districts and cities (shown in Figure 1), with a permanent population of 10.352 million (2019). The GDP amounts to 1159 billion Yuan, ranking 7th in the provincial capital cities of China.

Zhengzhou has a continental monsoon climate in the north temperate zone, with an average temperature of 15.6 °C for many years and an annual average precipitation of about 542.15 mm. The rivers in Zhengzhou belong to the Yellow River and Huai River, with a total amount of water resources of about 1.124 billion m^3^ and per capita water resource of about 179 m^3^. Moreover, the water resources are in relatively short supply and distribute unevenly in time and space. Zhengzhou plays an important role in the Chinese economy, responsible for economic development and societal progress in the central region. However, the increasing demand for water results in the shortage of water resources and the intensification of the contradiction between water supply and demand.

### 2.2. Data Resource

Herein, the data are from the statistical yearbooks, water resource bulletins, environment bulletins, government work reports of Zhengzhou from 2010 to 2019 and so on (limited by data update, the latest data available could only be obtained from 2019).

### 2.3. Research Methods

#### 2.3.1. Construction of Water Resource Carrying Capacity Index System

DPSIR is a conceptual model, proposed by the European Environment Agency after improving the PSR model. It systematically summarizes the interaction and relationship between the human and natural environment, as well as including some elements such as society, economy, resource and environment, characterized as comprehensiveness, scientific, systematisms, etc. [12,13,14,15]. In this model, water resources are divided into five subsystems, namely, driving force, pressure, state, impact and response. Among them, the driving force is a fundamental motivity to cause ecological environment change, and a potential inducement. Pressure can directly result in the ecological environment’s change. State is the reality of the ecological environment, induced by driving force and driven by pressure. Impact refers to the influence of the current situation of the ecosystem on the social economy, environmental state and human activity. Response refers to active measures and policies that deal with environmental problems. The basic idea of the DPSIR framework is concluded as follows. The driving factors in social and economic activities can lead to the environment’s change of state, mainly ascribed to their pressure. This process affects the social ecology, thus causing the policy response. Simultaneously, these countermeasures can also affect the driving factor, pressure, state and impact. Water resource carrying capacity is a complex system, which is affected by local economy, society, environment and so on. According to the sustainable development theory and the DPSIR model’s requirement, 19 indexes were selected to construct the evaluation index system of water resource carrying capacity for Zhengzhou based on the previous findings, as listed in Table 1. The data from 2010 to 2019 were used to analyze the temporal–spatial evolution characteristics of water resource carrying capacity of Zhengzhou.

#### 2.3.2. Index Weight Determination

The determination of index weight directly affects the accuracy and objectivity of the evaluation results. The existing expert investigation method, AHP method and subjective weighting method are simple and easy, but they ignore data influence and lack objectivity [16,17]. Different from the above methods, the entropy weight method (EWM) can directly use the existing indexes’ data to figure out their entropies according to the dispersion degree of the data, thus obtaining the weight values. This method effectively takes into account the variation degree of each index, so as to avoid the influence of the human factor to a certain extent and objectively reflect the importance level of the index [18,19]. Hence, this paper adopts a combination weighting method of the AHP method and EWM, that is, a combination of subjective weighting and objective weighting, to make the weight allocation more scientific and reasonable. The specific calculation steps are as follows (AHP formula is not listed).

(1)The original data are standardized(1)$$x_{ij^{\prime }}=\frac{x_{ij}-\min\left(x_{ij}\right)}{\max\left(x_{ij}\right)-\min\left(x_{ij}\right)}$$
where *i* = 1, 2, …, n; n is the evaluation index. *j* = 1, 2, …, n m; m is the year of evaluation. *x_ij_* is the initial value of index. *x_ij’_* is the value after standardization. The max (*x_ij_*) and min (*x_ij_*) are the initial maximum and minimum of corresponding indexes, respectively.
(2)The entropy *H_j_* of the index is calculated
(2)$$P_{ij}=\frac{x_{ij}}{\sum_{i=1}^{n}x_{ij}}$$
(3)$$H_{j}=-\ln{\left(n\right)}^{-1}\sum_{i=1}^{n}p_{ij}\ln p_{ij}$$
where *p_ij_* is the proportion of the *i*th evaluation index of the *j*th evaluation factor. *H_j_* is the entropy of the index, and $0\leq H_{j}\leq 1$. Obviously, when *P_ij_* = 0, *lnP_ij_* is meaningless. Thus, *P_ij_* should be revised.
(4)$$P_{ij}=\frac{1+x_{ij}}{\sum_{i=1}^{n}\left(1+x_{ij}\right)}$$
(3)The weight value of the evaluation index is calculated as follows
(5)$$W_{j}=\frac{1-H_{j}}{n-\sum_{j=1}^{n}\left(H_{j}\right)}$$
and their values meet $\sum_{j=1}^{n}W_{j}=1$, 0 < $W_{j}$ < 1.

Assuming that $W_{i1}$ and $W_{i2}$ are the weight values of the ith index, obtained by AHP method and EWM, respectively, the weight value $W_{i}$ of the *i*th index is calculated by the combination weighting method, as follows.
(6)$$W_{i}=\frac{W_{i1}W_{i2}}{\sum_{i=1}^{n}W_{i1}W_{i2}}$$

#### 2.3.3. Evaluation Model of Water Resources Carrying Capacity

TOPSIS method is also called the technique for order preference by similarity to ideal solution, which is used to sort the evaluation schemes from good to bad according to the Euclidean distance between the evaluation scheme and the ideal solution [20,21,22]. After analyzing the good and bad states of sample objects, this method establishes the optimal (worst) scheme, namely the positive (negative) ideal point, and determines the distance between the sample object and the positive (negative) ideal point. Thus, the closeness degree between the sample object and the ideal point is obtained. According to the basic principle of the TOPSIS method, it can avoid the subjectivity of data, as well as systematically analyze the gap between the current situation and the ideal state of water resource carrying capacity. Hence, the dynamic and change trend of the water resource carrying capacity of Zhengzhou is comprehensively and objectively reflected. The specific steps are as follows.

(1)Preprocessing of original data

The positive index and negative index are processed for the same trend. Here, the negative index evolves toward the trend of positive index. 

The calculation formulas are as follows.

Positive index is $y_{ij}\prime =y_{ij}$, and negative index is
(7)$$y_{ij}\prime =\frac{1}{y_{ij}}$$
where $y_{ij}$ is the initial value of index, *i* = 1, 2, …, n; *j* = 1, 2, …, m.

(2)Calculation of normalized matrix *Z*


(8)
$$Z_{ij}=\frac{y_{ij}^{}\prime }{\sqrt{\sum_{i=1}^{m}y_{ij}^{2}}}$$


(3)Construction of weighted normalized matrix *X*


(9)
$$X_{ij}=W_{i}\cdot Z_{ij}$$


(4)Determination of positive and negative ideal solutions

It is assumed that $X_{ij}^{+}$ is the maximum of the *i*th index in the year of *j*, that is, the optimal scheme, called the positive ideal solution. Likewise, the $X_{ij}^{-}$ is the minimum of the ith index in the year of *j*, that is, the worst scheme, called negative ideal solution. Their calculations are as follows.

The optimal scheme is
(10)$$X_{ij}^{+}=\left\{\left.\max X_{ij}\right|i=1,2,\cdots ,m\right\}=\left\{\left.\max X_{ij}\right|X_{i1}^{+},X_{i2}^{+},\cdots ,X_{im}^{+}\right\}$$

The worst scheme is
(11)$$X_{ij}^{-}=\left\{\left.\min X_{ij}\right|i=1,2,\cdots ,m\right\}=\left\{\left.\min X_{ij}\right|X_{i1}^{-},X_{i2}^{-},\cdots ,X_{im}^{-}\right\}$$

(5)Calculation of distances of the evaluation object from the positive ideal solution and negative ideal solution, respectively.

The distance of the evaluation object from the positive ideal solution is
(12)$$D_{ij}^{+}=\sqrt{\sum_{i=1}^{m}{\left(X_{ij}^{+}-X_{ij}\right)}^{2}}$$

The distance of the evaluation object from the negative ideal solution is
(13)$$D_{ij}^{-}=\sqrt{\sum_{i=1}^{m}{\left(X_{ij}^{-}-X_{ij}\right)}^{2}}$$

(6)Calculation of the closeness degree between the evaluation object and the ideal solution


(14)
$$T_{ij}=\frac{D_{ij}^{-}}{D_{ij}^{+}+D_{ij}^{-}}$$


The $T_{ij}$ value can reflect the closeness degree of the water resource carrying capacity to an optimal level. The larger the $T_{ij}$ is, the closer to the optimal level the water resource carrying capacity is. When $T_{ij}$ = 1, the water resource carrying capacity level is highest. However, when $T_{ij}$ = 0, the water resources carrying capacity level is lowest. In this paper, the closeness degree represents the level of water resource carrying capacity. The closeness degree of each year can be used to judge the level of water resource carrying capacity, thus judging whether it is good or bad. Based on the previous findings [23,24,25], the final results are divided into five grades according to the closeness degree $T_{ij}$ in a non-equidistant way in combination with the actual conditions of Zhengzhou. The specific grades are listed in Table 2.

## 3. Results and Analysis

### 3.1. Determination of Index Weight

According to the current water resource situation in Zhengzhou from 2010 to 2019, the comprehensive evaluation index system of water resource carrying capacity was established. Simultaneously, the AHP and EWM were used to calculate the subjective weight and objective weight, respectively. Finally, the combination weight W of the sustainable utilization evaluation index of water resources in Zhengzhou is obtained by Formulas (1)–(6), as shown in Table 3. 

### 3.2. Evaluation Results and Analysis of Water Resources Carrying Capacity of Zhengzhou

As can be seen from the analysis of comprehensive evaluation results of water resource carrying capacity of Zhengzhou, the comprehensive evaluation values of water resource carrying capacity increased from 0.4183 to 0.5560 between 2010 and 2019. Simultaneously, the carrying capacity upgraded from Grade III of normal carrying to Grade II of good carrying. Although there are small oscillations in some years, for example, the Grade IV mildly overloaded from 2012 to 2013, the water resource carrying capacity overall displays a fluctuating rise trend. The D+ value decreased from 0.0491 in 2010 to 0.0378 in 2019, indicating that the water resource carrying capacity gradually approached the positive ideal solution. Likewise, the D- value increased from 0.0353 in 2010 to 0.0473 in 2019, suggesting that the water resource carrying capacity gradually moved further away from the negative ideal solution.

In the sample period, the water resource carrying capacity was worst in 2012, with a T value of 0.3201. The comprehensive evaluation grade was Grade IV, between mild overloading grade and normal carrying grade. Notably, the water resource carrying capacity was optimal in 2018, with a T value of 0.5922. The comprehensive evaluation grade was GradeII, close to the critical value of good carrying grade. Obviously, the T value in 2018 is almost twice as high as the T value in 2012, implying that the water resource carrying capacity of Zhengzhou was significantly restored and improved during the sample period. Firstly, the original data were processed for the same trend using Formulas (7) and (8) to obtain the normalized matrix Z. Then, the comprehensive weight W of each index was substituted into Formula (9), thus obtaining the weighted normalized matrix. Finally, the comprehensive evaluation results of water resource carrying capacity for Zhengzhou from 2010 to 2019 were calculated using Formulas (10)–(14). The specific values and temporal–spatial evolution trends are displayed in Table 4 and Figure 2, respectively. The water resource carrying capacity reached Grade IV in 2012 and 2013, Grade III in 2010, 2011, 2014, 2015 and 2017, as well as Grade II in 2016, 2018 and 2019. The temporal–spatial change distributes in a trapezoid pattern, and steadily improves under ups and downs. The water resource carrying capacity level in 2018 and 2019 belong to the good carrying grade, but only to the critical level between Grade III and Grade II. Overall, the water resource carrying capacity level is lower, and belongs to the normal carrying grade. Therefore, the comprehensive carrying pressure is still severe.

Since 2012, Zhengzhou has increased efforts in ecological environmental governance, introduced high-tech industries, vigorously controlled the pollution of rivers, and eliminated backward production capacity with high water consumption. Moreover, the investment of water conservation projects increases year by year, and the efficient utilization of water resources is improved. The success of the Middle Route Scheme of the South–North Water Diversion Project alleviates the severe shortage of water to some extent, reduces the scale of groundwater exploitation in Zhengzhou, and basically meets the demand of economy development for water resources. Furthermore, large-scale governance and effective reform measures greatly improve the water ecological environment, water resource stock, water resource assets, and net assets of water resources, thus improving the level of water resource carrying capacity of Zhengzhou to a large extent. 

### 3.3. Evaluation Results and Analysis of Water Resources Carrying Capacity Subsystems of Zhengzhou

The sample data are substituted into the water resources carrying capacity model of Zhengzhou to work out the evaluation values of water resource carrying capacity subsystems, listed in Table 5. After the analysis of evaluation values, it was found that the change trends of water resource carrying capacity subsystems are basically consistent with those of the comprehensive level of water resource carrying capacity, shown in Figure 3. Moreover, the contribution of each subsystem to the comprehensive sustainable utilization level of water resource carrying capacity increases year by year, as displayed in Figure 4.

As can be seen from Figure 3, the carrying capacity index of the D subsystem overall decreased from 0.7095 in 2010 to 0.2076 in 2019, of which the carrying capacity index in 2011 shows the largest contribution, with a value of 0.7673. In terms of the component indexes of the D subsystem, natural population growth rate, urbanization rate and population density changed a little, but the growth rate of GDP changed obviously. As the economy slows down in China, the GDP of Zhengzhou is inevitably affected by the economy in China. During the sample period, the GDP maintained a medium-low growth rate of about 8%. Thus, the adjustment of industrial structure and the slowdown in urbanization speed resulted in the decrease in the D subsystem index of Zhengzhou.

Likewise, the carrying capacity index of the P subsystem shows a declining trend with fluctuation, from 0.4410 in 2010 to 0.2620 in 2019, except the large rebound in 2013 and 2016. From the component indexes of the P subsystem, the four indexes are all a negative index. In recent years, Zhengzhou has vigorously promoted water conservation, but some indexes, such as water consumption per capita, residents’ water consumption, the total amount of discharged sewage, and water consumption of industrial added value still continuously increase with migrant influx and economy transformation. This brings great pressure on the water resource carrying capacity of Zhengzhou.

Notably, the carrying capacity index of the S subsystem presents a stable rising trend with a small fluctuation. In 2010, the values of the S subsystem were 0.1524 and 0.8394 in 2019, approximately increasing by 550%. Therefore, the contribution of the S subsystem to the water resource carrying capacity was the highest in 2019. Among the component indexes, the sewage treatment rate and green coverage rate increased year by year, but the water consumption per 10,000 yuan of GDP decreased year by year. This is a direct driving force that keeps the contribution rate of the S subsystem continuously growing. On the contrary, the total amount of water consumption steadily increased, thus affecting the contribution of the S subsystem to a certain extent.

Furthermore, the carrying capacity index of the I subsystem sharply rose, and their values increased from 0.0169 in 2010 to 1 in 2019 in the sample period. It is suggested that the contribution of the I subsystem to the water resource carrying capacity largely increased, directly from the worst grade in 2010 to the optimal grade in 2019. During the sample period, Zhengzhou took a series of measures such as industry structure adjustment, industry upgrading, increase of the tertiary industry proportion, governance of water ecological environment, improvement of the water quality of rivers and lakes, as well as the implementation of water conservation measures. Accordingly, these measures can greatly improve the I subsystem’s index and increase the contribution of I subsystem in the whole system of water resource carrying capacity. Hence, they are considered as the core driving force to continuously improve the water resource carrying capacity in Zhengzhou.

The carrying capacity index of the R subsystem displays a fluctuating change. Its values rose from 0.4538 in 2010 to 0.5840 in 2019, increasing by 28.69% with a smaller amplitude. The results reveal that the contribution of the R subsystem to the water resource carrying capacity is stable, but the driving force is insufficient. Due to the influence of rainfall in Zhengzhou from 2010 to 2019, the fluctuation of the water resource stock index increased. The increase in the carrying capacity index of the R subsystem is mainly ascribed to the increasing investment of water conservancy projects in Zhengzhou year by year and the efficient utilization of water resource assets with rapid growth.

In conclusion, the D subsystem affected the development of water resource carrying capacity in Zhengzhou. Thus, water resources change and present different states. As such, the fluctuation range of the carrying capacity level of the P subsystem agrees well with that of the D subsystem, and changes with the change of the carrying capacity of the D subsystem. The S subsystem refers to the state of each subsystem, as well as the feedback to the human social environment. The I subsystem refers to the measures taken to deal with the problems of water resources, aiming to prevent the environment’s deterioration. It can be seen from the evaluation results that there is a greater difference in the carrying capacity change between the R subsystem and the other ones. This suggests that with economic and societal development, the governance and protection of the ecological environment in Zhengzhou still need to be strengthened.

## 4. Conclusions

Based on DPSIR theory framework and the TOPSIS model, water resources were evaluated through the comprehensive analysis of water resource carrying capacity level of Zhengzhou in full consideration of urban development. Our aim is to provide a decision-making basis for sustainable utilization, planning and management of water resources in Zhengzhou.

In general, the traditional water resources were evaluated using the water balance method. However, with the society and economy developing, human activity could intensify water resource evolution, thus resulting in the challenge of a technical approach. In this paper, the DPSIR framework was used to construct a comprehensive evaluation index system, including water resource, economy, society and environment, a total of four major elements. It not only reflected the influences of society and economy development as well as the human behaviors on the ecological environment, but also reflected the feedback of human behaviors and their resulting environmental state to society. The main conclusions of this study are as follows.

(1)From 2010 to 2019, the comprehensive evaluation values of water resource carrying capacity of Zhengzhou rose from 0.4183 to 0.5560, presenting a steadily increasing trend. Simultaneously, the grade of water resource carrying capacity improved from Grade III of normal carrying to Grade II of good carrying during the sample period. Among them, the lowest evaluation value of water resource carrying capacity was in 2012, i.e., 0.3201, whereas the highest evaluation value was in 2018, i.e., 0.5922. The latter is close to 185% of the former. Furthermore, the contribution of each subsystem to the comprehensive evaluation value increased year by year. Relatively, the contributions of the S subsystem and I subsystem to comprehensive carrying capacity were the greatest. The contribution of the R subsystem is stable. Affected by GDP growth rate and uneven temporal–spatial distribution of water resources in Zhengzhou, the carrying capacities of the D subsystem and P subsystem show the fluctuating change.(2)The combination weighting method was used to determine the index weight, which better avoids the influences of human subjective factors and data. Moreover, the difference between the status quo and the ideal value of water resource carrying capacity can be intuitively found by using the TOPSIS model. The evaluation results basically accord with the actual situation in Zhengzhou. Due to the data’s availability, this paper could only evaluate the water resource carrying capacity of Zhengzhou before 2019, which is the deficiency of this paper that needs to be improved in the future.(3)The water resource carrying capacity of Zhengzhou has significantly increased in recent years but is still at a general or lower level. In this regard, the sustainable utilization of water resources remains challenging. As a central city, Zhengzhou is experiencing a shortage of water resources, fragile water ecological environment, large population pressure, urgent upgrade of industrial structure, etc. Therefore, as the economy and urban construction are vigorously developed, some measures should be adopted, such as the reinforcement of residents’ awareness of water conservation, strengthening of the implementation of water ecological protection measures, improvement of the investment amount in the water conservancy project, introduction of high-tech industry, and a decrease in industrial water consumption, so as to further improve the water resource carrying capacity and promote the sustainable utilization of water resources.

## Figures and Tables

**Figure 1 ijerph-20-01394-f001:**
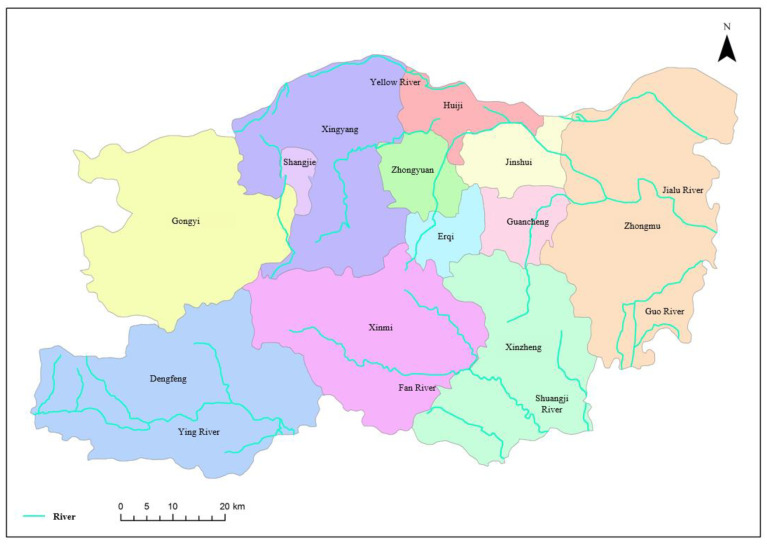
Administrative division of Zhengzhou city.

**Figure 2 ijerph-20-01394-f002:**
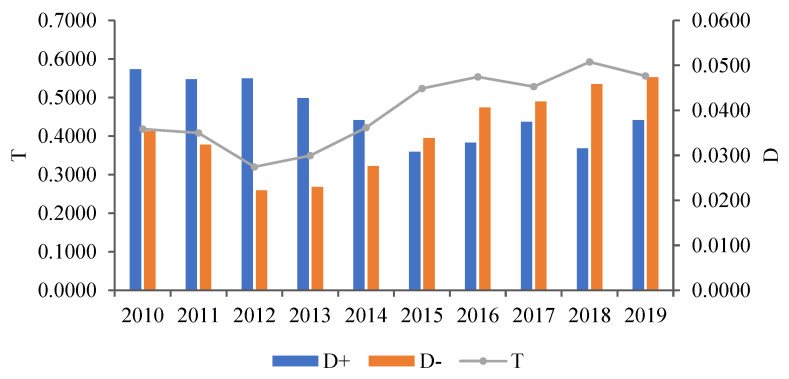
Change trend of comprehensive evaluation for water resource carrying capacity in Zhengzhou from 2010 to 2019.

**Figure 3 ijerph-20-01394-f003:**
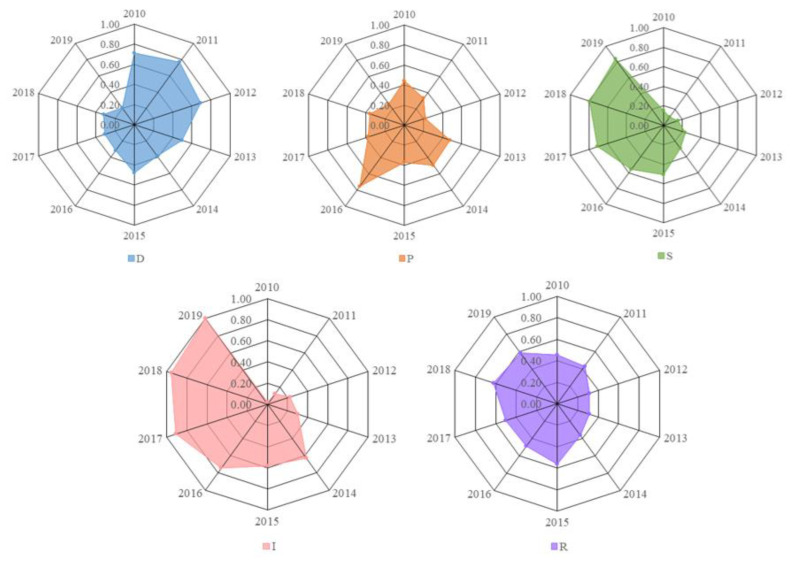
Change trend of subsystems of water resource carrying capacity in Zhengzhou from 2010 to 2019.

**Figure 4 ijerph-20-01394-f004:**
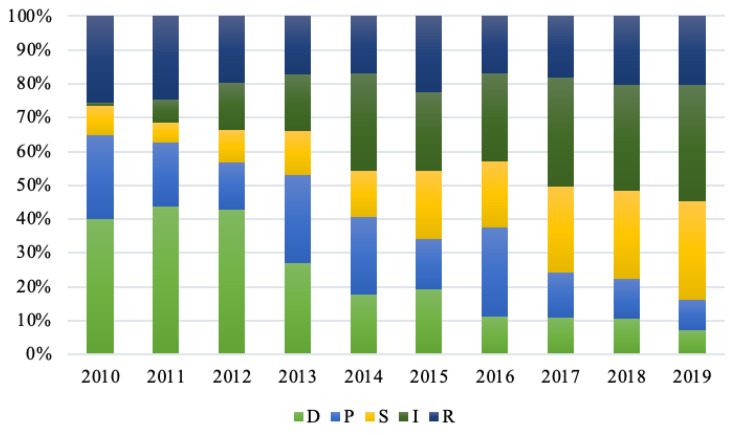
Contribution ratios of water resource carrying capacity subsystems of Zhengzhou from 2010 to 2019.

**Table 1 ijerph-20-01394-t001:** Evaluation index system of water resource carrying capacity in Zhengzhou based on DPSIR model.

Evaluation Factor	Evaluation Index	Unit	Index Attribute	Index Code
Driving force (D)	GDP growth rate	%	+	D_1_
	Natural population growth rate	‰	+	D_2_
	Urbanization rate	%	+	D_3_
	Population density	Persons/km^2^	+	D_4_
Pressure (P)	Water consumption per capita	m^3^	−	P_1_
	Residents’ water consumption	100 million m^3^	−	P_2_
	The total amount of discharged sewage	100 million m^3^	−	P_3_
	Water consumption of industrial added value	m^3^/ten thousand yuan	−	P_4_
State (S)	Sewage treatment rate	%	+	S_1_
	Total water consumption	100 million m^3^	−	S_2_
	Green coverage rate	%	+	S_3_
	Water consumption per 10,000 yuan of GDP	m^3^/ten thousand yuan	−	S_4_
Impact (I)	Water quality compliance rate of water sources	%	+	I_1_
	Water quality compliance rate of sections	%	+	I_2_
	Per capita disposal income	Ten thousand yuan	+	I_3_
	Proportion of tertiary industry	%	+	I_4_
Response (R)	Water resource stock	100 million m^3^	+	R_1_
	Water resource assets	100 million yuan	+	R_2_
	Investment of water conservancy project	100 million yuan	+	R_3_

**Table 2 ijerph-20-01394-t002:** Grading standards of water resource carrying capacity for Zhengzhou.

Grade	Grade V	Grade IV	Grade III	Grade II	Grade I
Grade description	Serious overloading	Mild overloading	Normal carrying	Good carrying	Surplus carrying
$T_{ij}$	(0–0.25]	(0.25–0.35]	(0.35–0.55]	(0.55–0.75]	(0.75–1]

**Table 3 ijerph-20-01394-t003:** Comprehensive weight results of the index system.

Evaluation Factor	Index Code	AHP Weight (W_1_)	Entropy Weight (W_2_)	Combination Weight(W)
D	D_1_	0.3317	0.2852	0.0726
(0.2004)	D_2_	0.1972	0.2267	0.0343
	D_3_	0.3317	0.2801	0.0713
	D_4_	0.1394	0.2081	0.0223
P	P_1_	0.2865	0.2860	0.0848
(0.2663)	P_2_	0.2026	0.2186	0.0458
	P_3_	0.3407	0.2745	0.0968
	P_4_	0.1703	0.2209	0.0389
S	S_1_	0.1409	0.3274	0.0230
(0.1212)	S_2_	0.4105	0.2280	0.0467
	S_3_	0.1273	0.2039	0.0129
	S_4_	0.3212	0.2406	0.0386
I	I_1_	0.2053	0.2368	0.0297
(0.1534)	I_2_	0.3453	0.2656	0.0560
	I_3_	0.2053	0.2692	0.0337
	I_4_	0.2441	0.2284	0.0340
R	R_1_	0.4934	0.3477	0.1327
(0.2587)	R_2_	0.3108	0.3054	0.0734
	R_3_	0.1958	0.3469	0.0525

**Table 4 ijerph-20-01394-t004:** Comprehensive evaluation values of water resource carrying.

Year	D+	D−	T
2010	0.0491	0.0353	0.4183
2011	0.0469	0.0324	0.4083
2012	0.0471	0.0222	0.3201
2013	0.0427	0.0230	0.3494
2014	0.0378	0.0276	0.4217
2015	0.0308	0.0339	0.5236
2016	0.0328	0.0406	0.5534
2017	0.0375	0.0419	0.5282
2018	0.0316	0.0458	0.5922
2019	0.0378	0.0473	0.5560

**Table 5 ijerph-20-01394-t005:** Evaluation values of water resource carrying capacity subsystems of Zhengzhou from 2010 to 2019.

Year	D	P	S	I	R
2010	0.7095	0.4410	0.1524	0.0169	0.4538
2011	0.7673	0.3296	0.1026	0.1213	0.4323
2012	0.6884	0.2227	0.1536	0.2238	0.3140
2013	0.4934	0.4794	0.2368	0.3051	0.3150
2014	0.3848	0.4915	0.2947	0.6267	0.3655
2015	0.4795	0.3692	0.5061	0.5824	0.5600
2016	0.3163	0.7538	0.5606	0.7429	0.4841
2017	0.3060	0.3763	0.7073	0.9083	0.5044
2018	0.3224	0.3602	0.7966	0.9548	0.6212
2019	0.2076	0.2620	0.8394	1.0000	0.5840

## Data Availability

The raw data supporting the conclusions of this article will be made available by the authors without undue reservation.

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
