# Peer review of "Study on Water Resource Carrying Capacity of Zhengzhou City Based on DPSIR Model"

_ijerph, 2023, doi:10.3390/ijerph20021394_

Round 1

Reviewer 1 Report (Previous Reviewer 1)

I agree with the revision made by the authors. I have no other suggestions.

Author Response

Thanks to the reviewer#1 for your acknowledgments and suggestions. But this opinion is not aimed at this article due to the pasting error of the editorial department. After communication and reflection with the editor, the editorial department has re attached the correct revision comments. Therefore, we did not make any targeted revisions to the comments in Review#1. We hereby explain.

Reviewer 2 Report (Previous Reviewer 2)

I looked at the revised manuscript and the author's answers to the questions. I think that the manuscript has been significantly improved. The modifications made increased the informative value of it and in my opinion it gives an interesting perspective on the matter.

Author Response

Thank you very much for your approval and suggestions. In response to your comments, we have carefully revised the article, improved the sentences and expressions in the article, revised the tables and figures in the article, and recalculated the accuracy of the data calculation process and results (line10-13;60-65;113-114;213;225;243;301;334-335;392-393).Thank you very much for your comments.

This manuscript is a resubmission of an earlier submission. The following is a list of the peer review reports and author responses from that submission.

Round 1

Reviewer 1 Report

In this study, the authors uses the DPSIR model to study the water resources carrying capacity of Zhengzhou. There are some problems in this study:

 1. How does the authors define the DPSIR model, and what is the connotation of these elements for water resource carrying capacity? What is the Driving force? What is Pressure? What is State?   What is Impact? What is Response? The authors needs to express clearly in the manuscript.

 2. At present, many scholars have done a lot of research on water resources carrying capacity. Where is the innovation of the author's research in this MS?

 3. Water resources carrying capacity includes production, domestic and ecology. How does the authors reflect this?

 4. How to verify the conclusions of this study? Is the result reliable? What is the contribution of each indicator to the water resources carrying capacity? Are there regional differences?

Reviewer 2 Report

An interesting view on the assessment of water resources. It gives an opportunity to think about the reality, what all can affect them. At the same time, it is important to understand where and which factor to include within DPSIR. Overall, I have a very positive opinion on the contribution.

Some remarks:

The geographical names of the streams should be unified, I would recommend the original language and the translation into English only if used or known

Fig. 1, add the names of the streams

2.2

Probably add the type of data, or maybe just link to the tab. 1if is relevant

2.3.2, 2.3.3

I recommend, due to the number of formulas and indexes, to change the formatting of the text so that it is better visible what a paragraph is, what is a heading, formula, explanations, etc., it visually blends a little

Table 2: better highlight the boundaries of the intervals for individual Grades. And my question is why the indicated intervals do not visually follow each other, if they were listed in the opposite order it would be more transparent (my opinion)

Table 1 and Table 3:  to unify the evaluation factor (P) indexing